# Protein Interactome Profiling of Stable Molecular Complexes in Biomaterial Lysate

**DOI:** 10.3390/ijms232415697

**Published:** 2022-12-10

**Authors:** Yuri Mezentsev, Pavel Ershov, Evgeniy Yablokov, Leonid Kaluzhskiy, Konstantin Kupriyanov, Oksana Gnedenko, Alexis Ivanov

**Affiliations:** Institute of Biomedical Chemistry (IBMC), 119121 Moscow, Russia

**Keywords:** interactomics, protein–protein interactions, mass-spectrometric identification, size-exclusion chromatography, interactome taxonomy of proteins, stable protein complexes, molecular fishing, affine chromatography

## Abstract

Most proteins function as part of various complexes, forming via stable and dynamic protein–protein interactions (PPIs). The profiling of PPIs expands the fundamental knowledge about the structures, functions, and regulation patterns of protein complexes and intracellular molecular machineries. Protein interactomics aims at solving three main tasks: (1) identification of protein partners and parts of complex intracellular structures; (2) analysis of PPIs parameters (affinity, molecular-recognition specificity, kinetic rate constants, and thermodynamic-parameters determination); (3) the study of the functional role of novel PPIs. The purpose of this work is to update the current state and prospects of multi-omics approaches to profiling of proteins involved in the formation of stable complexes. Methodological paradigm includes a development of protein-extraction and -separation techniques from tissues or cellular lysates and subsequent identification of proteins using mass-spectrometry analysis. In addition, some aspects of authors’ experimental platforms, based on high-performance size-exclusion chromatography, procedures of molecular fishing, and protein identification, as well as the possibilities of interactomic taxonomy of each protein, are discussed.

## 1. Introduction

Most of the protein molecules in living systems function not alone but as part of various protein complexes—from the simplest dimeric structures to complex molecular systems consisting of dozens of subunits. The formation of protein complexes is realized by protein–protein interactions (PPI). Therefore, the scientific field called protein interactomics is aimed at solving three main problems: (1) identification of protein partners and parts of complex intracellular structures; (2) establishing the parameters of PPI (affinity, molecular-recognition specificity, rates of complexes’ formation and dissociation, thermodynamics, and complex structure); (3) study of the functional role of the identified PPI. Successful solutions of these problems can significantly expand our fundamental knowledge about the structure, principles of functioning, and regulation of protein complexes and intracellular molecular machineries. In terms of application, it is important to find PPIs that can be used as targets for innovative drugs that selectively regulate biological processes at the level of PPIs.

The initial task of protein interactomics is the systemic analysis of PPI and cataloging of protein complexes. The complexity of this task is due to the multilevel architecture, structural and functional dynamism, and flexibility of the protein interactome as a result of molecular rearrangements in the cell as a response to the action of many endogenous and exogenous factors.

Diverse protein complexes can be characterized using a number of criteria: (1) the number of individual proteins (subunits) involved in the complexes—dimers, trimers, tetramers, etc.; (2) subunit identity—homo- and hetero-dimers, etc.; (3) functional significance—mono- or multi-functional complexes; (4) the number of proteins interacting with the target protein in the oligomeric complex.

Functionally significant protein complexes can be conventionally divided into stable (long-lived) and dynamic (short-lived, also known as metastable or transient) ones. This assessment of protein complexes is based on the following PPI parameters: equilibrium dissociation constant (K_D_), association and dissociation rate constants (k_on_ and k_off_, respectively), and half-life (t_1/2_) of a complex [1,2,3]. Dynamic protein complexes are typically multifunctional because they are part of many signaling and metabolic pathways, either through the presence of multiple domains or by the formation of transient complexes that vary in function [4]. Moreover, the lifetime of a dynamic protein complex determines the effectiveness of a certain biological process.

Databases such as hu.MAP v.2.0 [5] (accessed on 5 September 2022, http://humap2.proteincomplexes.org/), CORUM v.3.1 [6] (accessed on 5 September 2022, https://mips.helmholtz-muenchen.de/corum/), and Complex Portal (accessed on 5 September 2022) [7] (https://www.ebi.ac.uk/complexportal/home) are important web-based tools for cataloging and interpreting interactome-profiling data and contain information on 6965, 5134, and 2472 protein complexes, respectively. However, one of the main problems of protein interactomics is the lack of information about the involvement of most proteins in the formation of stable protein complexes. This problem is further complicated by the fact that the spectrum of PPIs of a particular protein may differ depending on its subcellular and tissue localization, as well as in normal conditions and in the presence of a pathological process [8]. There is still a general methodological problem of analyzing and distinguishing between stable and dynamic protein complexes [9]. It is due to the lack of unified protocols for the preparation of tissue or cellular lysates that results in the different dissociation rate of native protein complexes and an existence of many customized techniques for protein complexes’ separation. The purpose of this work is to assess the current possibilities and prospects of interactome profiling of proteins involved in the formation of stable protein complexes based on the methodological paradigm of separation of proteins and their stable complexes with subsequent mass-spectrometric identification.

## 2. Interactome Profiling of Stable Protein Complexes

In the field of protein interactomics, there is still no universal method for PPI detection, so most studies use various combinations of proteomic methods. Each of them is optimized for highly specialized tasks with its own set of advantages and disadvantages. A panel of several state-of-the-art analytical methods is often used to identify possible stable protein complexes. These include yeast two-hybrid assay [10], the chemical cross-linking approach [11,12], density-gradient ultracentrifugation [13], co-immunoprecipitation, and tandem affinity purification [14,15,16], as well as blue native gel electrophoresis [17,18].

All methods of protein co-elution are based on the separation of protein complexes under native conditions. The basic idea here is that proteins belonging to the same complex elute or migrate together during separation [19]. Among these methods, co-fractionation based on size-exclusion-chromatography (SEC) fractionation [20,21,22] and ion-exchange-chromatography (IEX) fractionation [23,24] are most commonly used due to their high reproducibility of routine experimental protocols and availability of scientific equipment. The high technical level of interactome profiling based on SEC and/or IEX in conjunction with other proteomic methods allows the achievement of good separation of individual proteins and protein complexes [25,26,27]. For an accurate panoramic assessment of the distribution of native proteins between monomeric and oligomeric forms, as well as heterocomplexes in lysate fractions, there is an approach combining SEC fractionation with multi-angle light scattering (MALS) [28,29]. For semi-quantitative and quantitative evaluations of proteins in lysate samples, different variants of mass-spectrometric identification of proteins are used [30,31], including density-gradient analysis by mass spectrometry (qDGMS) [32], stable-isotope labeling with amino acids in cell culture (SILAC) [33,34], and pulse SILAC (pSILAC) to study the dynamics and turnover number of proteins in complexes [35]. The general scheme of panoramic protein interactome profiling can be divided into two blocks: “Data-generation block” and “Analytical block” (Figure 1). The data-generation block consists of preanalytical and preparative phases. The analytical block includes phases of bioinformatics and verification of the results of interactome profiling.

### 2.1. Preanalytical and Preparative Phases

The preanalytical stage consists of choosing and optimizing the method of sample preparation of a biological sample for the conversion of protein material into a lysate using mild solubilization conditions in order to preserve the native stable protein complexes. Depending on the goals of the experiment, it is possible to include the stage of sample enrichment with a material with a specific subcellular localization, for example, a cytoplasmic or membrane fraction.

The preparative step includes either SEC fractionation of the lysate material or serial rounds of SEC and IEX in order to increase the separation efficiency of protein complexes with different physicochemical properties. The principle of SEC is to separate the components of a complex mixture according to their molecular weights (whereas IEX uses their charge differences) in a separation column packed with chemically inert sorbent. The larger the molecule, the smaller the depth of its penetration into the sorbent granule containing pores with different sizes. As a result, the path of large molecules in the sorbent is shorter than that of small molecules. Therefore, large molecules leave the separation column faster than small ones. The principle of high-performance SEC is to separate the lysate material into multiple fractions, each containing proteins and multiprotein complexes of the same molecular weight (MW). As a result of subsequent analysis of the protein content in the fractions, protein elution profiles can be obtained depending on the properties of the “molecular sieve” used. To do this, after SEC fractionation, proteins in each fraction are subjected to enzymatic proteolytic cleavage (for example, trypsinolysis) according to standard protocols, followed by mass-spectrometric analysis (LC-MS/MS) of proteolytic peptides and bioinformatics data processing.

A typical example of a chromatogram of a high-performance SEC fractionation of a whole tissue lysate is shown in Figure 2. Data were obtained from our previous experiments [36]. The tissue lysate sample was separated into 22 fractions, and for comparative analysis of the protein composition of the fractions, six representative fractions were chosen. These fractions covered the range of single-protein molecules (45–60 kDa), the range of dimers and trimers (60–150 kDa), and the range corresponding to high-molecular-weight protein complexes (150–450 kDa). With an average protein molecular weight of 42 kDa (according to the UniProt database), the latter complexes consisted of at least four or more subunits. It follows from the chromatogram that there are two pronounced protein peaks, one of which is in the region of single-protein molecules (MW ≈ 45 kDa), and the other peak ≈ 350 kDa, whose area is 2.5 times larger than the former, is in the region of protein complexes with MW range from 150 to 350 kDa.

Further, the distribution matrix of individual proteins, identified by LC-MS/MS analysis in the test lysate fractions covering the range from 15 to 700 kDa, is presented in Table 1. It shows that, in most fractions, there are many proteins with significantly lower MW than the average MW of the fraction, determined by the calibration of the chromatographic column. These proteins, obviously, could appear in these SEC fractions only as part of protein complexes with MW corresponding to the average MW of the fractions. It should be noted that in rare cases, the presence of proteins with MW above the average MW of the fraction is observed. This effect may be due to the presence of proteolytic fragments of these proteins or high-adhesion properties of some proteins, which causes their slow movement in the SEC column.

Formally, three main types of protein states can be distinguished according to SEC elution profiles: (I) the protein is present only in the monomeric form and does not participate in the formation of stable protein complexes, (II) the protein is present in the form of homodimers and/or heterodimers, (III) the protein participates in the formation of multimeric complexes of a higher order. Intermediate types of protein states (IV–VI) include different combinations of types I, II, and III. Table 2 provides examples of such a conditional classification based on SEC fractionation of whole liver tissue lysate and mass-spectrometric protein identification [36]. For example, apolipoprotein A–I (31 kDa) is found both in monomeric form and in SEC fractions with MW values of 250 and 440 kDa as part of high-order complexes. Therefore, the identification of several types of protein states makes it possible to obtain specific information about the participation of each protein in the formation of stable protein complexes and the sizes of these complexes. In addition, combinatorial analysis of the MW sums of identified proteins in a SEC fraction allows us to compile a limited list of potential protein partners for each target protein.

### 2.2. Bioinformatic Phase

The generation of hypotheses on the composition of stable protein complexes is often based on the correlation of protein co-elution profiles. The tissue and cellular lysates used for SEC fractionation contain a heterogeneous mixture of thousands of individual proteins and protein complexes with different compositions. In order to predict the compositions of possible stable protein complexes present in each fraction, a number of bioinformatic algorithms are used to analyze the LS-MS/MS data [30,37]. These algorithms use certain combinatorial assumptions and data clustering, which generate the redundant number of final hypotheses, especially in the case of predicting the protein heterocomplexes containing three or more subunits. In order to reduce the redundancy of hypotheses, systemic biological filters are used, such as gene co-expression, data on known protein complexes and paired PPI (for example, the CORUM portal), gene ontology analysis, and PPI networks construction [38]. The prediction of combinatorial hypotheses of binary protein complexes (both homodimers and heterodimers), which can be rather easily verified experimentally in vitro, can serve as an additional tool for the analysis of protein co-elution profiles. In the absence or difficulty in obtaining protein co-elution profiles (for example, when analyzing proteins in only one SEC fraction), we offer our own “Dimers” program, the code of which is presented in a Figshare repository (DOI: 10.6084/m9.figshare.21526443). One of the possible algorithms used in our practice for tissue-lysate SEC-fraction PPI data interpretation is presented in Figure 3. As can be seen from the figure, the two-stage data analysis “Dimers” algorithm includes the generation of combinatorial hypotheses for dimeric protein complexes and the selection of hypotheses by two descriptors: MW and a semi-quantitative assessment of the protein content in the fraction. At the first stage, a combinatorial analysis of protein pairs is carried out so that the sum of their MW (molecular weight of a possible dimer) is in the range of ± 10% of the average MW of the SEC fraction. Intermediate list #1 contains a list of predicted dimeric protein complexes that meet this criterion. For example, in the case of an average MW of the SEC fraction equal to 45 kDa, this interval corresponds to the range from 40.5 to 49.5 kDa. At the second stage, for each pair of proteins (conditional dimer) from list #1, a semi-quantitative assessment of their content in the fraction is performed according to the values of emPAI (exponentially modified protein abundance index) [39]). This approach is based on the assumption that both proteins are involved in the formation of only joint dimeric complexes, and accordingly, these values for two interacting proteins should not be significantly different. If protein X is involved in the formation of two dimeric complexes XY and XZ, then its emPAI value may differ significantly from proteins Y and Z, provided that the latter are involved in the formation of only these complexes. So, if the quotient of dividing the larger emPAI value (protein X) by the smaller value (protein Y) does not exceed 1.5, then this possible dimer is registered in list #2. Thus, the final list #2 contains only those dimeric protein complexes from list #1 that consist of a pair of proteins with close emPAI values. For example, if the emPAI values for protein X and for protein Z are 1.4 and 1.0, respectively, then the division quotient is 1.4 (which is less than 1.5); therefore, the two proteins X and Z can hypothetically participate in the formation of a dimeric complex. Usually approximately 10% of predicted combinatorial protein complexes from list #1 end up in list #2, which significantly reduces the total number of PPI hypotheses. Hypotheses of target protein complexes (list #2) can first be theoretically “verified” by analyzing pre-existing interactomics information in the electronic databases or by constructing PPI networks by means of methods of mathematical modelling [40,41]. Thus, by using the STRINGdb portal (accessed on 12 September 2022, https://string-db.org/), one can explore data on the mutual positioning of two proteins in the PPI network (the presence of direct interactions and common protein partners, as well as the shortest paths between them). It is also advisable to include additionally in the general protocol the computer modeling of possible 3D structures of the proposed dimeric complexes using the methods of molecular docking [42] and molecular dynamics [43], including the AlphaFold2 and RoseTTAFold platforms [44,45]. In the case of a positive result of theoretical “verification” of a potential dimeric protein complex, its experimental verification can be planned using one or more PPI-detection methods [46].

### 2.3. Experimental Verification of Possible Protein Complexes

A variety of methods can be used to verify stable protein complexes identified by panoramic interactome profiling of proteins in tissue and cellular lysates. One approach that we have successfully used to identify protein partners that form stable complexes with a target protein is the direct-molecular-fishing procedure. The protocol of this procedure is based on the affinity isolation and mass-spectrometric identification of potential protein partners from the whole lysate (or from its SEC fractions), where a target protein (“bait” protein) is covalently immobilized on a solid support (chromatography resins or paramagnetic nanoparticles) [47,48,49]. However, it should be noted that the molecular-fishing procedure allows us to isolate from the lysate not only the first-order protein partners that directly interact with the target protein but also second-order, third-order, etc. that are part of the complex structure. Therefore, for proteins that were found in the same SEC fraction with a target protein and were also isolated by the molecular-fishing procedure on an affinity sorbent with the same target protein, it is desirable to perform instrumental verification of their PPI to differentiate direct and indirect protein partners. Among the existing various proteomic technologies that are suitable for PPI detection, we preferentially use surface plasmon resonance (SPR) optical biosensors, where the first protein is immobilized on the surface of a special optical chip and the second protein is injected into the biosensor flow cell in different concentrations. PPIs are recorded in real time as a set of sensorgrams that can be used to calculate the rate constants’ association and dissociation, as well as the equilibrium dissociation constant (K_D_) of protein complex.

As an example of the implementation of an approach, based on the intersection of SEC profiling and molecular-fishing data, interactomic analysis of microsomal cytochrome b5 (CYB5A) as a target protein can be demonstrated. Previously, a semi-quantitative assessment of the distribution of proteins co-fractionated with CYB5A in SEC fractions of liver tissue lysate was performed [36]. We found that, among them, there are eight metabolic enzymes (CAT, LSS, UGP2, CYP4A11, ACAA1, EPHX1, ALDH1A1, and CYP2C9), which were isolated from lysates prepared from liver tissue and HepG2 cells, in molecular-fishing experiments using CYB5A as a target protein [36,50]. Protein-distribution profiles in the SEC fractions of the lysate are shown in Figure 4. It can be seen from the figure that almost all the amount of ACAAT1 (MW = 44.8 kDa) is present in monomeric form in the SEC fraction (45 kDa), while CYB5A (MW = 15.3 kDa) is present in trace amounts in this SEC fraction. This indicates the absence of stable protein complexes between these proteins, and one can only assume that ACAAT1 is an indirect partner of CYB5A. Control SPR experiments showed a positive concentration-dependent binding of microsomal cytochrome P450 2C9 (CYP2C9) with covalently immobilized CYB5A on the optical chip of a biosensor [50]. CYP2C9 and CYP4A11 are considered to be known functionally significant direct protein partners of CYB5A presented in high-MW SEC fractions. The ALDH1A1 protein (MW = 55.4 kDa) exists as a homotetramer [51]. However, its peak content is found in SEC fractions from 100 to 170 kDa, which corresponds to its dimeric form, which is co-fractionated with the CAT (MW = 60 kDa) and CYB5A (MW = 15.3 kDa) proteins. The formation of a heterocomplex involving CYB5A–CAT–EPHX1–ALDH1A1 is possible in the region of MW ≈ 200 kDa, while in the region of higher-molecular-weight complexes (250–450 kDa), the co-fractionation of CYB5A, EPHX1 (MW = 53.1 kDa), and UGP2 (MW = 57 kDa) takes place. Thus, in this example, out of eight of CYB5A’s protein partners, isolated by the molecular-fishing procedure, it can be assumed that only three proteins (CAT, EPHX1, and UGP2) are direct-interaction partners for CYB5A, according to a high similarity of their co-fractionation profiles.

## 3. Biomedical Aspects of Protein Interactome Profiling

Determination of the spectrum of disease-associated proteins and their mapping in PPI networks is important for understanding the mechanisms of pathogenesis. It is also useful for finding ways to influence key molecular points, the functioning of which is impaired both in infectious and non-infectious diseases [52]. This research area has been successfully developing along with the advances in protein-identification technologies, new experimental omics approaches, and bioinformatics software for data interpretation [53,54]. Thus, the omics approach based on SEC and label-free quantitative MS profiling made it possible to catalog stable nuclear protein complexes isolated from the human glioblastoma cell line T98G with partial verification by an independent method of direct interactions of a number of proteins co-fractionated with histone deacetylase 1 (HDAC1) [55]. According to the literature, certain PPIs are known that can be directly related to the etiology of human diseases (examples of which are given in Table 3), mainly with neoplastic transformation as the most-studied phenomenon at the molecular level. A broad spectrum of disease-related PPIs favors the use of protein-interactome-profiling methods, including SEC-LC-MS/MS, to identify such PPIs with the prospect of studying their medical significance for disease prognosis, as well as finding PPIs as molecular targets for pharmacotherapy and gene therapy.

Preparative SEC-based separation of proteins in lysates followed by LC-MS/MS analysis was used to detect quantitative changes in the elution profile of proteins in HT29 intestinal adenocarcinoma cell culture when treated with the HSP90 inhibitor tanespimycin [73]. New HSP90-dependent PPIs involving the isocitrate dehydrogenase [NAD] regulatory subunit 3 (IDH3) were also identified. To reveal on- and off-targets for pharmacologically active small molecule compounds in tissue or cellular lysates, a number of modern proteomic approaches based on the detection of binding, such as drug affinity responsive target stability (DARTS) and cellular thermal shift assay (CETSA) and its variants—thermal proteome profiling (TPP) and isothermal dose–response CETSA [74,75,76], are used. Along with activity-based protein profiling (ABPP) [77] utilizing chemical proteomics technologies for detecting labeled molecules, the SEC-LC-MS/MS protocol can be implemented as an additional panoramic method for mapping potential protein targets for the action of biologically active metabolites or drugs. It should be noted that the SEC-LC-MS/MS protocol, unlike ABPP, provides important information about stable protein complexes sensitive to the presence of an unmodified metabolite, while irreversible interactions of chemically modified compounds in the active center of one of the proteins can greatly complicate the search for modulators of the target PPI. Comparison of the results of protein profiling of the lysate pre-incubated in the absence (control) or presence of biologically active compounds/drugs reveals changes in the co-elution profiles of some proteins involved in the formation of stable protein complexes. Such “heat” PPIs can be considered as potential targets for active compounds or drugs. Previously, we studied the effect of the endogenous non-peptide bioregulator isatin (indole-2,3-dione) on SEC profiles of proteins in rat liver tissue lysate using the SEC-LC-MS/MS protocol [78]. Lysate samples were incubated on ice for one hour with the addition of working buffer (control) and isatin solution at a final concentration of 100 μM (experiment), which corresponds to the limit of its physiological concentration in tissues. Comparative analysis of protein SEC profiles revealed that the addition of isatin affected the SEC profiles of about half of the identified known isatin-binding proteins. Moreover, we found three new isatin-dependent proteins whose oligomeric state is modulated by isatin: ATP-citrate synthase (tetramer), triosephosphate isomerase (dimer), and argininosuccinate synthase (tetramer). In addition, analysis of the profiles of known isatin-binding proteins showed that there are 20 proteins that participate in the formation of the stable part of the protein interactome. Among them, about 65% of the identified proteins participate in the formation of multimeric protein complexes, 25% in homodimers or heterodimers, and only 10% are detected as single molecules.

## 4. Conclusions

Studies of stable protein complexes encounter an extremely important methodological aspect consisting of the transformation of a biomaterial (tissue samples or cells) into a liquid phase (lysate). Therefore, the methods of sample preparation of biomaterial for the analysis of the protein interactome as a whole and the preparation of tissue lysates, in particular, carry elements of customization. This is especially true for the choice of the composition of lysis buffers, since there is no standardization in these approaches that are focused on the stabilization of the specific native protein complexes. The existence of a large number of lysis buffers and protocols for the preparation of lysates increases the objective differences in the results of interactome profiling, which complicates the comparative analysis of data from different sources.

Another problematic aspect of protein interactome studies is the participation of not only canonical forms of proteins in PPIs but also their various proteoforms arising as a result of alternative splicing and post-translational modifications. Moreover, the appearance of specific post-translational modifications of protein partners is possible, which occur only after the formation of stable complexes. To date, there is no systematic information on the relationship between the formation of protein complexes and the presence of specific proteoforms [79], which requires more detailed investigations.

In conclusion, it should be noted that there is a positive outlook for the deeper fundamental understanding of protein-interactome-profiling data. This is an opportunity to carry out partial interactome taxonomy of each identified protein. The participation of the protein in the formation of stable complexes (the stable part of the interactome) can be characterized by two groups of parameters. The first group (qualitative parameters) includes the following: (1) the fact that the protein participates in the formation of stable protein complexes; (2) the fact that the protein is involved in one or more complexes; (3) the size of the complexes in which the protein participates; (4) the fact that the protein can be in the free monomeric state; (5) a list of proteins found in the same fractions along with the analyzed protein (co-fractionation hypotheses); (6) a list of potential protein partners (in case of the sum of the molecular weights of the partner protein and the target protein corresponds to the average MW of the SEC fraction); (7) hypotheses about the possible functional role of the protein. The second group includes the following semi-quantitative parameters: (1) the percentage distribution of the protein between the fractions (monomeric form and complexes of different sizes), (2) the ratio of the amount of each protein in the SEC fraction to the amount of the analyzed protein, (3) the list of potential protein partners based on semi-quantitative estimates.

## Figures and Tables

**Figure 1 ijms-23-15697-f001:**
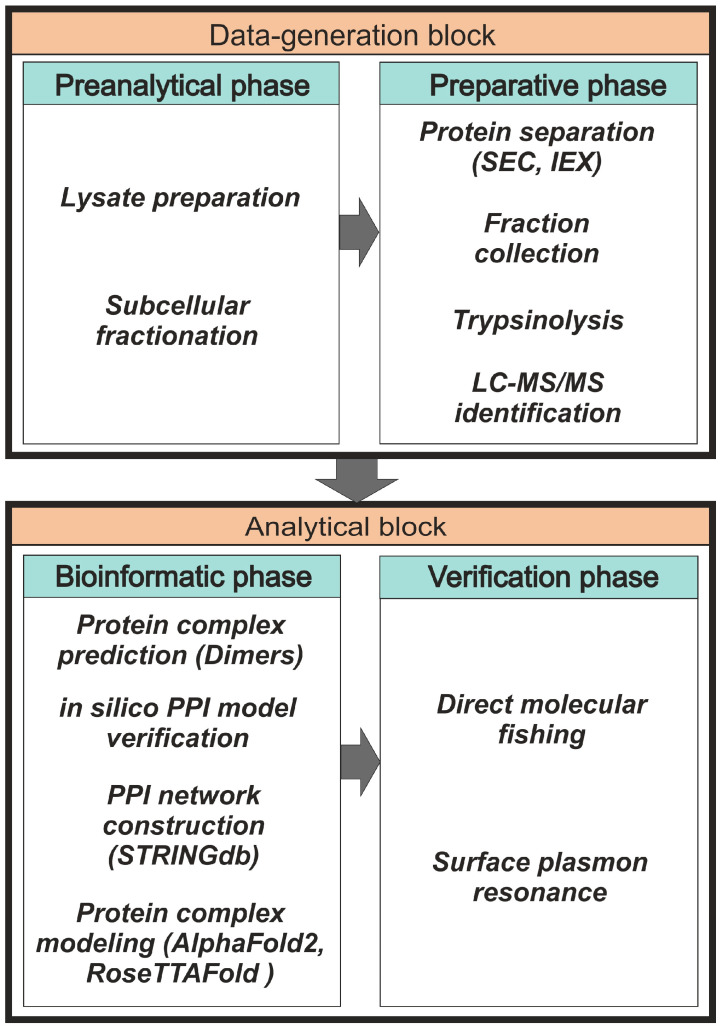
The general scheme of the panoramic-interactome-profiling protocol.

**Figure 2 ijms-23-15697-f002:**
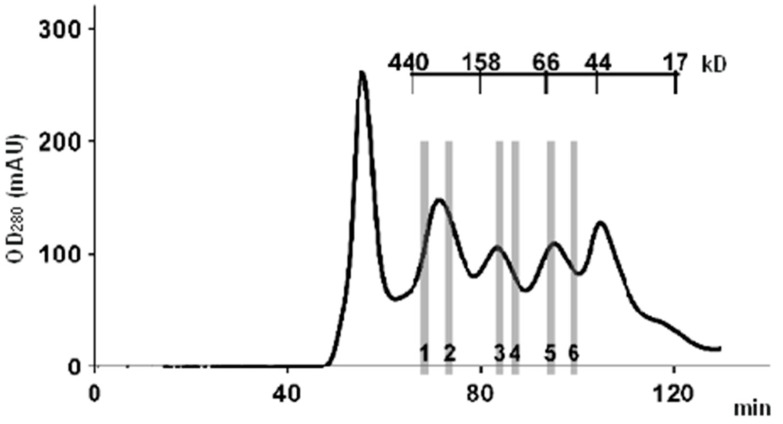
A typical chromatogram of a high-performance SEC fractionation of whole liver tissue lysate. AKTA Purifier 10 chromatograph (GE Healthcare) and HiLoad 16/600 high resolution column (GE Healthcare), filled with Superdex 200 and equilibrated with HBS-EP buffer (10 mM HEPES (pH 7.4), 150 mM NaCl, 3 mM EDTA, and 0.05% Tween-20), were used for protein separation. Six lysate fractions, which were selected for comparative analysis of the protein composition, are indicated by numbers 1–6. The figure was adapted with permission from Ref. [36]; “Pleiades Publishing, Ltd.”, 2018.

**Figure 3 ijms-23-15697-f003:**
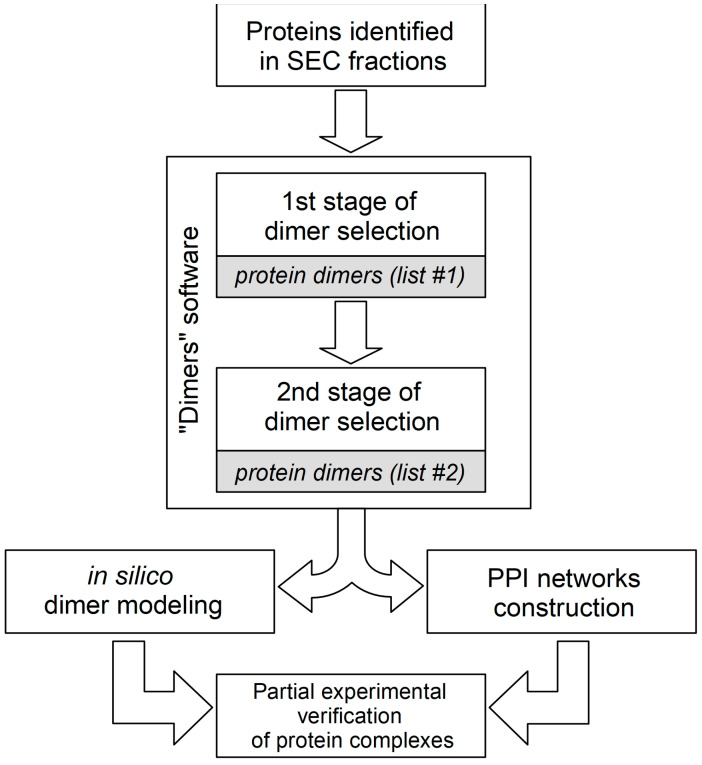
Algorithm for interpreting data on protein–protein dimers in fractions obtained by SEC separation. It consists of two stages of generating dimer hypotheses in the “Dimers” program; as a result, two lists of hypotheses are formed. There is also a selection of the most prioritized hypotheses by in silico methods for further experimental verification.

**Figure 4 ijms-23-15697-f004:**
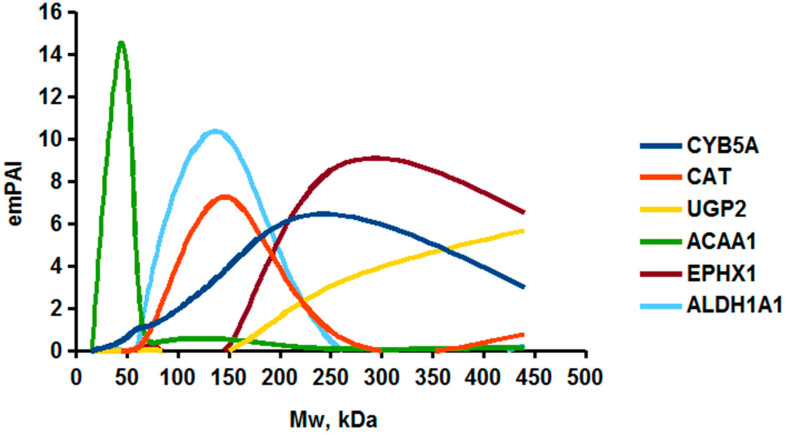
Co-fractionation profiles of CYB5A and its protein partners in the liver tissue lysate (according to the data in [36,50]). The profiles were plotted using points, which correspond to average molecular-weight values of SEC-fraction MW: 15, 30, 48, 60, 90, 130, 290, and 400 kDa. The *x*-axis shows the MW of the SEC fractions, and the *y*-axis shows the semi-quantitative assessment of a protein content in each SEC fraction. LC-MS/MS analysis was performed on an Orbitrap Fusion hybrid orbital mass spectrometer (Thermo Scientific) in the positive ionization mode in an NSI (nanospray ion) source (Thermo Scientific). The selection of proteins had Mascot score values > 50. The following are the abbreviated protein names: CYB5A—microsomal cytochrome b5 (UniProt ID P00167); CAT—catalase (UniProt ID P04040); UGP2—UTP-glucose-1-phosphate uridylyltransferase (UniProt ID Q16851); ACAA1—3-ketoacyl-CoA thiolase, peroxisomal (UniProt ID P09110); EPHX1—epoxide hydrolase 1 (UniProt ID P07099); and ALDH1A1—aldehyde dehydrogenase 1A1 (UniProt ID P00352).

**Table 1 ijms-23-15697-t001:** A distribution matrix of proteins identified by LC-MS/MS analysis in SEC fractions of liver tissue lysate.

MW (kDa) of Identified Proteins	Mean MW OF Fraction (kDa)					
	400	290	130	90	60	48
<11	5	12	1	4	4	2
11–21	33	42	37	18	13	25
21–31	86	75	53	45	34	80
31–41	117	110	55	56	68	84
41–51	93	106	56	54	70	78
51–61	77	88	43	53	59	65
61–71	29	32	25	28	28	29
71–81	22	24	14	14	22	20
81–91	12	10	4	3	7	3
91–101	10	9	10	7	4	5
101–111	11	11	6	9	5	5
111–121	10	3	4	3	5	0
121–131	7	4	5	7	3	4
131–141	4	4	0	0	0	0
141–151	6	5	1	1	1	0
151–176	4	2	1	1	1	1
176–201	4	5	3	1	0	0
201–226	0	0	1	0	1	1
226–251	0	0	0	2	2	1
251–301	8	4	2	1	1	1
301–351	0	0	0	0	0	0
351–401	0	0	0	0	1	0
401–451	0	0	0	0	0	0
451–501	0	0	0	0	0	0
501–601	1	1	0	0	1	1
501–701	1	1	1	1	0	1
Total	540	548	322	307	330	406

Proteins with MW values less than the average MW of a fraction are highlighted in green. Proteins whose MW values correspond to the average MW of a fraction are single proteins and highlighted in yellow. Proteins whose MW values exceed the average MW of a fraction are likely fragmented or with high-adhesion properties and highlighted in light-gray color. The data were obtained in the framework of [36] and are presented here for the first time).

**Table 2 ijms-23-15697-t002:** Types of conditions of proteins in tissue lysates based on SEC elution profiles.

Type	Description	Examples *	Protein Name
I	Monomeric form only **	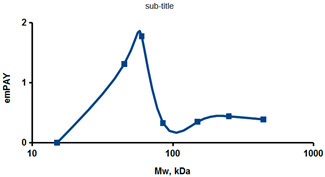	Apoptosis-inducing factor 1 (mitochondrial), O95831, 66.9 kDa, monomer
II	Homodimers and heterodimers	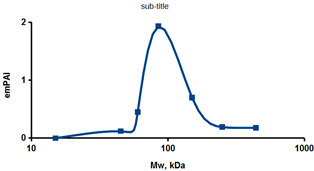	3-hydroxyisobutyrate dehydrogenase (mitochondrial), P31937, 35.3 kDa, homodimer
III	Homooligomers and heterooligomers ***	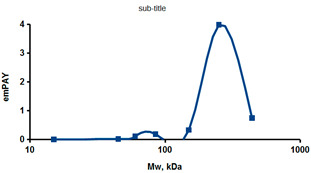	Very long chain specific acyl-CoA dehydrogenase (mitochondrial), P49748, 70.4 kDa, homodimer
IV	I and II	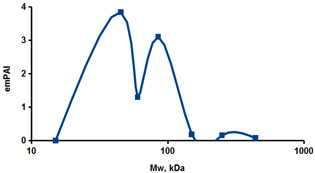	Aspartate aminotransferase (mitochondrial), P00505, 47.5 kDa, homodimer
V	I and III	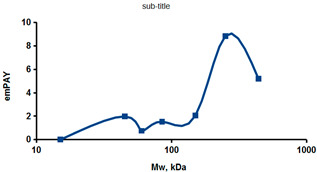	Apolipoprotein A–I, P02647, 30.8 kDa, homodimer
VI	I, II, and III	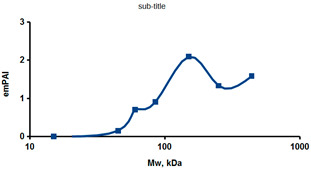	Peroxisomal multifunctional enzyme type 2, P51659, 79.7 kDa, homodimer

* the dots from left to right indicate the following MW: 15, 45, 60, 85, 150, 250, and 440 kDa. ** MW of a protein corresponds to average MW of a SEC fraction. *** the number of subunits > 2. The data were obtained in the framework of [36] and are presented here for the first time.

**Table 3 ijms-23-15697-t003:** Examples of disease-related PPIs.

Protein Complex	Disease	Prognostic Value	Drug Target
LMO2/LDB1 *, LMO2/LDB1/TAL1/E12	Cancer [56,57]	[58,59]	[60]
TP53/EP300	Cancer [61]	[62]	[63]
FGFx/FGFRx	Cancer [64]	[65]	[66]
TP63/mutTP53	Cancer [67]	[68]	[69]
HTT/HAP-1	Neurological disorders [70]	[71]	[72]

* LMO2—nuclear LIM domain only 2; LDB1—LIM domain-binding protein 1; TAL1—T-cell acute leukemia protein 1; E12- E-protein 12; TP53—cellular tumor antigen p53; EP300—histone acetyltransferase p300; TP63—tumor protein 63; FGF (R)—fibroblast growth factor (receptor); HTT—huntingtin; HAP-1—huntingtin-associated protein-1; mut—mutant.

## Data Availability

A program code, deposited at the Figshare repository (DOI: 10.6084/m9.figshare.21526443), will be available after the article is published.

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
