# Peer review of "Protein Interactome Profiling of Stable Molecular Complexes in Biomaterial Lysate"

_ijms, 2022, doi:10.3390/ijms232415697_

Round 1

Reviewer 1 Report

The author presents a pipeline to update the current approaches for protein interactome profiling. The first divides the work in few steps: 1) prenanlysis phase to choose and optimize the method sampling of a protein complex/unit and preparative phase which increase the separation efficiency of protein complexes using different physico-chemical properties, 2) Bioinformatic algorithms employed to analyze the MS data using different biological filters. 3) experimental verification using molecular fishing procedure to identify the partner proteins in a stable complex. The authors ran experiments to ran and differentiate direct and indirect protein partners. 

Overall the organization of the paper is good. However, the description of the Bioinformatic phase could be more elaborate. At present, the reference listed as list #1, list#2 is understandable. The procedure used needs more description and may be an overall figure of the pipeline of the work would make it more readable. Figure #1 could be more elaborate, at this point this is very basic.

The Figure #3 mentions about grey highlight which is not present in the figure.  

Author Response

The author presents a pipeline to update the current approaches for protein interactome profiling. The first divides the work in few steps: 1) prenanlysis phase to choose and optimize the method sampling of a protein complex/unit and preparative phase which increase the separation efficiency of protein complexes using different physico-chemical properties, 2) Bioinformatic algorithms employed to analyze the MS data using different biological filters. 3) experimental verification using molecular fishing procedure to identify the partner proteins in a stable complex. The authors ran experiments to ran and differentiate direct and indirect protein partners.

Response:

We really appreciate your help in improving the presentation of the work. We have tried to take into account all your comments and recommendations for our work as much as possible. We have made all the relevant edits and additions, and also made some corrections to improve the work grammatically from the point of view of the English language.

Overall the organization of the paper is good. However, the description of the Bioinformatic phase could be more elaborate. At present, the reference listed as list #1, list#2 is understandable. The procedure used needs more description and may be an overall figure of the pipeline of the work would make it more readable. Figure #1 could be more elaborate, at this point this is very basic.

1) The Fig. 1 was changed, the Bioinformatic phase was broadened to be more readable. In the text new sentences related to the modified figure 1 were added:

“The general scheme of panoramic protein interactome profiling can be divided into two blocks: “Data generation block” and “Analytical block” (Fig 1). Data generation block consists of preanalytical and preparative phases. Analytical block includes phases of bioinformatics and verification of the results of interactome profiling.”

2) The following text elaborating the list #1 and list #2 was added in section 2.2. Bioinformatic phase:

“Intermediate list #1 contains a list of predicted dimeric protein complexes that meet this criterion.”

“Thus, the final list #2 contains only those dimeric protein complexes from list #1 that consist of a pair of proteins with close emPAI values.”

“Usually approximately 10% of predicted combinatorial protein complexes from list #1 end up in list #2, which significantly reduces the total number of PPI hypotheses.”

3) The Fig. 5 containing elaboration of bioinformatic data analysis including list #1 and #2 generation was added to the manuscript with the corresponding title.

The following text was added to the manuscript:

“One of the possible algorithms used in our practice for tissue lysate SEC-fraction PPI data interpretation is presented in figure 5. As can be seen from the figure,”

The Figure #3 mentions about grey highlight which is not present in the figure.

In figure 3, we highlight three areas: green, yellow and light gray. Perhaps there was a misunderstanding. We called “light gray” as “gray”. The Figure 3 title was corrected accordingly.

Reviewer 2 Report

The paper is a well-done review on protein-protein interactions, including also practical aspets. The field is important and the Authors provided a really up-to-date survey of the broad literature (it should be emphasized, that of the 79 references 12 are from 2022!). The paper needs, however some English and scientific-style amendment (including also grammar).

A few lines on the working principles of Size Exclusion Chromatography would be interesting.

Specific remarks:

"Journalistic" style expressions should be avoided. Outstanding is the frequent use of "profiling" - starting from the title, then e. g. in rows 12, 18, 62, 73, 76, 90, 231, 336 - maybe that there are still some more. 

row 16: thermodynamics => thermodynamic

row 20 author's ... it appears that there are more than one authors, thus authors' is correct

row 30: perhaps: function not alone...

rows 33-36 structure of the intermolecular complexes is also an important target

row 124: "Data" is plural, thus => data have been...

rows 124/125: "previous experiments" reference/references is/are lacking

row 146 and several other places: Mw should be better written as MW

Table 1 - reference is lacking 

row 184: content => composition

Figure 3 is a kind of table, while Table 1 is a kind of figure. Is this necessary?

Reference 73 is unusual. Is this a book? or article? 

Author Response

The paper is a well-done review on protein-protein interactions, including also practical aspets. The field is important and the Authors provided a really up-to-date survey of the broad literature (it should be emphasized, that of the 79 references 12 are from 2022!). The paper needs, however some English and scientific-style amendment (including also grammar).

Response:

We really appreciate your help in improving the presentation of the work. We have tried to take into account all your comments and recommendations for our work as much as possible. We have made all the relevant edits and additions, and also made some corrections to improve the work grammatically from the point of view of the English language.

A few lines on the working principles of Size Exclusion Chromatography would be interesting.

The following paragraph was added to the manuscript text:

“The principle of SEC is to separate the components of a complex mixture according to their molecular weight (whereas IEX uses their charge differences) in a separation column packed with chemically inert sorbent. The larger the molecule, the smaller the depth of its penetration into the sorbent granule containing pores of a different size. As a result, the path of large molecules in the sorbent is shorter than that of small molecules. Therefore, large molecules leave the separation column faster than small ones.”

"Journalistic" style expressions should be avoided. Outstanding is the frequent use of "profiling" - starting from the title, then e. g. in rows 12, 18, 62, 73, 76, 90, 231, 336 - maybe that there are still some more.

Authors suggest that “protein profiling” is rather the actual scientific term in the field of proteomics (DOI: 10.1007/978-1-59745-281-6_30) than the popular science jargon. Therefore, with all due respect, authors would appreciate it if this term was not removed from the text.

row 16: thermodynamics => thermodynamic

Corrected accordingly.

row 20 author's ... it appears that there are more than one authors, thus authors' is correct

Corrected accordingly.

row 30: perhaps: function not alone..

Corrected accordingly.

rows 33-36 structure of the intermolecular complexes is also an important target

Corrected accordingly.

row 124: "Data" is plural, thus => data have been…

Corrected accordingly.

rows 124/125: "previous experiments" reference/references is/are lacking

Corrected accordingly.

row 146 and several other places: Mw should be better written as MW

Corrected accordingly.

Table 1 - reference is lacking

The Table 1 footer was changed to

 *the dots from left to right indicate the following MW: 15, 45, 60, 85, 150, 250 and 440 kDa.

**MW of a protein corresponds to average MW of a SEC-fraction

***the number of subunits > 2

The data were obtained in the framework of [36] and presented here for the first time.

row 184: content => composition

Corrected accordingly.

Figure 3 is a kind of table, while Table 1 is a kind of figure. Is this necessary?

Table’s formatting rules of IJMS Microsoft Word template do not imply a clear separation of cells and highlighting them with color, so the authors considered it optimal to present the data in the form of a picture.

Table 1 has rows and columns that match the structure of the table. The authors considered it possible to present them in the form of a table.

Reference 73 is unusual. Is this a book? or article?

Thank you for your valuable commentary. We have found a misprint in the references list. Reference 73 is an article preprint. The reference was changed accordingly to the following:

Samant, R.S.; Batista, S.; Larance, M.; Ozer, B.; Milton, C.I.; Bludau, I.; Biggins, L.; Andrews, S.; Hervieu, A.; Johnston, H.E.; et al. Native Size Exclusion Chromatography-Based Mass Spectrometry (SEC-MS) Identifies Novel Components of the Heat Shock Protein 90-Dependent Proteome; Biorxiv, 2022, 1-28.